# Transcending Conventional Biometry Frontiers: Diffusive Dynamics PPG Biometry

**DOI:** 10.3390/s21165661

**Published:** 2021-08-23

**Authors:** Javier de Pedro-Carracedo, David Fuentes-Jimenez, Ana María Ugena, Ana Pilar Gonzalez-Marcos

**Affiliations:** 1Departamento de Tecnología Fotónica y Bioingeniería, ETSI Telecomunicación, Universidad Politécnica de Madrid (UPM), E-28040 Madrid, Spain; javier.depedro@uah.es; 2Departamento de Electrónica, Universidad de Alcalá (UAH), Escuela Politécnica Superior, Alcalá de Henares (Madrid), E-28871 Alcalá de Henares, Spain; david.fuentes@depeca.uah.es; 3Departamento de Matemática Aplicada a las Tecnologías de la Información, ETSI Telecomunicación, Universidad Politécnica de Madrid (UPM), E-28040 Madrid, Spain; anamaria.ugena@upm.es

**Keywords:** biometric system, PPG signal dynamic, 0–1 test, CNN architecture, pattern analysis

## Abstract

This paper presents the first photoplethysmographic (PPG) signal dynamic-based biometric authentication system with a Siamese convolutional neural network (CNN). Our method extracts the PPG signal’s biometric characteristics from its diffusive dynamics, characterized by geometric patterns in the (p,q)-planes specific to the 0–1 test. PPG signal diffusive dynamics are strongly dependent on the vascular bed’s biostructure, unique to each individual. The dynamic characteristics of the PPG signal are more stable over time than its morphological features, particularly in the presence of psychosomatic conditions. Besides its robustness, our biometric method is anti-spoofing, given the complex nature of the blood network. Our proposal trains using a national research study database with 40 real-world PPG signals measured with commercial equipment. Biometric system results for input data, raw and preprocessed, are studied and compared with eight primary biometric methods related to PPG, achieving the best equal error rate (ERR) and processing times with a single attempt, among all of them.

## 1. Introduction

The relentless outbreak of the pandemic in our lives has put the globalized world in check. The paralysis to which economies across the globe are driven has been reversed, in many cases, by the spread of a latent wave for decades: the digitization of society. Life will be conditioned by new technologies, an entire online ecosystem whose real impact remains a chimera even among those experts who timidly venture into hasty forecasts [1,2,3].

The role that technology will play in future societies is unquestionable. However, this profound metamorphosis carries challenges that digital platforms themselves have to face. One of them is to keep the identities of the users of the different services protected, that is, to avoid identity theft so that the platform can unequivocally verify that a user is who they say they are and not an impostor intruder with clearly fraudulent purposes. Today, the most secure authentication mechanisms are based on biometric methods [4]. Compared to traditional access passwords, the different biometric identification systems are reliable and free the user from memorizing numerous keys [5]. The only access password lies in the user’s anatomical characteristic, supposedly exclusive and non-transferable, whose emulation is extremely problematic even for the most seasoned intruders. Face, voice, iris, palm, and finger recognition are already a reality that safeguards socioeconomic transactions [6,7,8].

The conventional biometric systems focus on the analysis of physical characteristics of an individual, in some cases highly sensitive to involuntary morphological disturbances—for example, a cut on the fingertip undergoes a fingerprint analysis. By contrast, biological signals lend themselves to a more robust biometric examination. Besides morphological details of the biological signal waveform, dynamic peculiarities by the expected functional response of the physiological system of interest are evaluated.

In recent decades, the preliminary diagnostic examination of an individual’s state of health and its follow-up has been entrusted on many occasions to clinical analysis, through non-invasive methods, of the biological signals generated by the human body—more recently, with body sensor networks (BSN) and thanks to the rapid development of health informatics [9,10]. Among the different biological signals usually measured today, one particularly deserves special consideration, the photoplethysmographic (PPG) signal [11,12].

Since Alrick Hertzman, an American physiologist, devised the first photoelectric photoplethysmograph in 1937 [13], although rudimentary, recent technological advances provide devices, such as modern pulse oximeters, that are increasingly smaller, lighter, and with a marked tendency to market themselves as wireless devices at a very affordable price [14,15]. An essential aspect of the PPG technique lies in its low sensitivity to the sensors’ location, which gives versatility to photoplethysmography for its application in many areas, such as health, sports, or the agri-food industry. Its appearance has been due to the electronic simplicity, the cost-benefit ratio, the ease of signal acquisition, and, mainly, its non-invasive character [16,17,18]. Unlike other biological signals that require bulky measurement equipment, or even accessories, such as gels (EEG) or electrodes (ECG), the PPG signal requires relatively modest electronics. Uncomplicated electronics and optoelectronics encourage the construction of small pulse oximeters, easily integrable into smart devices [19]. A pulse oximeter consists of a light emitter and a photodetector. The photodetector senses changes in light absorption resulting from arterial blood pulses (pulse signal or PPG) when a light beam passes through or reflects in human tissue [20].

The PPG signal is widely used in clinical settings to monitor physiological parameters related to the cardiorespiratory system [21]. It is complex. It is composed of an AC component—peripheral pulse synchronizes to each heartbeat—and a quasi-DC part that varies slowly due to respiration, vasomotor activity, and vasoconstrictor waves [22]. The mutual coupling between the different components is intricate and operates at different timescales to regulate blood volume based on physiological needs.

### 1.1. PPG Biometric System—State of the Art

The development of biometrics during the 20th century—according to its definition in [23]: “Measurable physical characteristics or personal behavioral traits used to identify or verify the identity of an individual”—began by conforming to the old paradigm of facial recognition and fingerprints. Nevertheless, continued progress in the area of image processing and analysis has fostered the exploration of more sophisticated biometric system designs [24,25] (for a known review of classical biometric approaches and their evolution over time, readers are referred to [26], and for advanced deep learning technologies in biometric recognition to [27]).

So far, in the 21st century, the development of biometric pattern recognition systems has evolved enormously, broadening its application spectrum in the context of morphological analysis, as reflected between the proposal of the anatomical characterization of the hand geometry in [8] and that made by [28] concerning 3D palmprint modeling. The same is true for other biostructure patterns as disparate as the geometric characterization of the ear [29], the iris [30], the eye as a multimodal biometric system [31], face detection [32], of the distribution of veins in a finger [33] or on the wrist [34], and of 3D fingerprint identification [35].

However, in this century, particular attention must be paid to the use of biological signals such as biometric markers, in addition to morphological and behavioral characteristics. In this regard, it is worth highlighting biometrics studies involving the analysis of electrocardiographic (ECG) and encephalographic (EEG) signals [36], to which could be added biometric applications that obtain biological signals from galvanic response of skin (GSR), electromyogram (EMG) [10], electrooculography (EOG), and mechanomyogram (MMG), among others [37].

Over the years, technological advances have simplified the acquisition of biological data; somehow, *traditional biometric systems* (TBS) have been increasingly giving way to *wearable biometric systems* (WBS) and, thus, to new methodological approaches to computing and validating biometric patterns [38]. Accordingly, new biometric technologies are gradually abandoning the rigidity imposed by a stationary and static analysis of biometric patterns [39] towards biometric patterns adapted to the variations that the biological signals may undergo over time—the so-called *adaptive biometric systems* [40]. In the particular case of the PPG signal, biometric patterns are strongly conditioned to physiological alterations, such as physical activity, emotional states, and time intervals, in which measurements will do, apart from the impact of the different noise sources coupled in the PPG signal acquisition procedure [19], mainly when the PPG signal is obtained from a camera or of wrist-worn PPG collected in an ambulant environment [41].

Focusing now on the matter at hand, the first documented reference to the PPG-based biometric system dates back to Gu et al.’s research in 2003 [42]. In all the works that use the PPG signal as a biometric reference, specific biomarkers correspond to features implicitly or explicitly extracted from the signal waveform: for example, time-domain features acquired from the PPG signal’s first and second derivatives for biometric identification [43], approximating each PPG signal as a sum of Gaussians and using the parameters in a discriminant analysis framework to distinguish individuals [44], or defining the waveform of the PPG signal in five consecutive PPG cycles [45], from 22 cycles [46] or 100 cycles [47] parametrically. One of the latest works is related to the non-fiducial and fiducial approaches for feature extraction with supervised and unsupervised machine learning classification techniques [48], recently expanded with other multi-feature classification techniques [49,50]. Another is the simultaneous PPG signal acquisition using different wavelengths that allows the video camera detectors to extract the colour segment (e.g., red, green, and blue) [51]. In all PPG-based biometric models, a negative aspect is the non-stationary nature of the PPG signal over time, which prevents the stable identification of an individual’s biometric patterns.

### 1.2. PPG Biometric System—Proposal

This work uses the PPG signal dynamics as a biometric reference of any individual. In this sense, we focus our attention on the geometric distribution of the PPG signal’s diffusive behavior, according to the (p,q)-plane proposed by the 0–1 test [52,53,54]. We feel that the PPG signal’s diffusive dynamics are unique to each individual since the diffusion constant of blood flow is subject to the structural configuration with which each individual has been endowed [55]. A whole complex network of arterioles and capillaries transports blood from the heart to the rest of the body thanks to the heart’s driving force and being synchronized with the respiratory rhythm. Although variations in the PPG signal’s diffusive dynamics can hide point or progressive pathological abnormalities, such as physiological deterioration resulting from aging, specific congenital characteristics remain practically unchanged.

Each subject’s credentials and identity are collected in blood flow dynamics through the peripheral capillary network. Its falsification is very difficult because of the capillary network’s intricacy and the complexity which involves blood flow driven by the cardiorespiratory system. Furthermore, a significant detail is that any biometric system verifying the PPG signal’s diffusive dynamics requires the individual’s vital integrity. Someone, not without a negligible effort, could imitate the particular capillary morphology of an artificial finger. Still, it would be practically impossible to reproduce the diffusive dynamics that blood flow undergoes when circulating through that capillary structure, given the contribution of many subsystems that nonlinearly make up the cardiovascular system.

The paper is organized as follows. Section 2 describes the two fundamental concepts that are applied for the first time on biometry. The mathematical framework of the 0–1 test, which underpins the biometric potential of the geometric patterns traced by the PPG signal’s diffusive behavior, is in Section 2.1, and Section 2.2 explains our novel proposal for a biometric classifier based on convolutional neural networks in detail. Section 3 is about the data, optimizer, and logic error employed in the experiment; it includes a brief description of the parameters used to evaluate the system. Section 4 shows the obtained results, both graphically and numerically, for various experimental settings. Additionally, in this section, we analyze and interpret the obtained results. Finally, in Section 5, we shortly outline the conclusions drawn from this study, which serve as the basis for future work.

## 2. Method

In PPG-based biometrics within the deep neural network (DNN) framework, as a general concept of the system, we propose a biometric system based on the diffusive dynamics of the PPG signal with a DNN design adapted to diffusive images and a specific biometrics method. Our proposal technically rests on the 0–1 test [56] and the Siamese residual network structures.

### 2.1. 0–1 Test

In analyzing dynamical systems, one of the key aspects is to characterize the dynamic behavior present in the physical system’s response under study. The response dynamics do not provide direct relevant information on the internal physical structure from which the response derives. Still, it does provide at least its operational complexity, which is crucial in evaluating its correct functioning and its greater or lesser adaptability to unforeseen situations in the context of physiological systems.

In an experimental setting, observables are usually obtained from the physical system under consideration so that the observables are making measurements at regular time intervals. An observable is any physical quantity that can be measured. The measurements or observations themselves are in what is known as time sequences (time series), and then each observable gives rise to a scalar time sequence (scalar time series).

We could define a state vector in phase space if we measured all the observables contributing to a given dynamical system evolution. In physiological systems, it is widespread to work with univariate time series or scalar time series, in which only the measurements of an observable are available. With a single observable, it is possible to obtain information on the system’s state since each usually contains information from the others, given the mutual coupling between them, whether linear or non-linear.

The 0–1 test’s initial motivation was to have a method applied directly to a scalar time series to identify the presence of chaotic dynamics without resorting to other, more complicated techniques requiring a deep level of knowledge for its correct application and interpretation [52,53,57]. Given its easy implementation, its increasing popularity has sparked the interest of countless scientific disciplines in an excessive race to detect chaos anywhere [54]. However, beyond the initial scope of the 0–1 test and its many applications, one of the steps of the test is surprisingly useful in the field of biometrics: specifically, the auxiliary trajectory of the two-dimensional Euclidean group (*the Fourier transform series*), or *p*-*q* diagram or (p,q)-plane [56], which underlies the dynamics of the physical system.

The 0–1 test cornerstone construction of an extended dynamic serves a two-dimensional Euclidean group SE(2) [56]. The elements of SE(2) form rigid displacements, that is, a translation and a rotation, in some two-dimensional affine Euclidean plane—the (p,q)-plane—that, in principle, does not relate in topological terms to the state space in which the dynamics of the system unfold. However, parameters that characterize rigid transformations depend at all times on the current state of the system. Therefore, there is a certain equivalence relationship itself between the dynamics of the physical system under study and the dynamic evolution of the trajectory described by the elements of SE(2) in the (p,q)-plane.

The 0–1 test requires as input a scalar time series of *N* observations s(n), for n=1,2,⋯,N, where s(n) is a one-dimensional observable of the underlying dynamical system. According to the rigid transformations’ parameterization, the extension of the dynamics is characterized by s(n) forces to define three scalar quantities (p,q,ϕ). An element or point on the (p,q)-plane is defined by its position on the plane, whose coordinates are (p,q), although its evolution, a change in coordinates, is driven (*forcing term*) by the dynamic evolution of s(n) according to
(1)pn+1=pn+s(n)cosϕn,qn+1=qn+s(n)sinϕn,ϕn+1=c+αs(n),
where parameters c,α∈R.

The evolution of any point on the (p,q)-plane describes a trajectory called the auxiliary trajectory since it reproduces an indirect or complementary evolution of the true dynamics observed in the system. The auxiliary trajectory involves an angular rotation ϕn with respect to a circumference of radius s(n) centered on the point (pn,qn), as shown in Figure 1.

Somehow, the auxiliary trajectory derives from a diffusive process in which the diffusion dynamics are forced or driven by the s(n) observations. In the presence of noise, for dynamic simplicity, α usually assigns a value of 0 [53,57] so that Equation (Equation 1) is reformulated as
(2)pn=∑k=1ns(k)cos(kc),qn=∑k=1ns(k)sin(kc),ϕn=cn,
where the angle of rotation ϕn increases at a uniform rate governed by the value of *c*. Furthermore, since the parameter *c* participates in the trigonometric function’s argument, it is pertinent that c∈[0,2π).

Although the theory underlying the dynamic extensions is based on the dynamics’ asymptotic behavior, an interesting consequence focuses on the limited nature of auxiliary trajectories in the (p,q)-plane, that is, how the auxiliary trajectory evolves spatially in the (p,q)-plane if the trajectory is circumscribed in an area delimited or inexorably diffuses in the same way that a Brownian motion unfolds [57]. The 0–1 test quantifies, by the computation of an indicator, whether the auxiliary trajectory is bounded. It reflects the presence of regular dynamics or those not sublinearly bounded, which manifests chaotic dynamics. This inductive argument is the basis of the 0–1 test; a more in-depth description goes beyond this paper’s purpose. Readers are referred to this method’s original work, widely referred to in the scientific literature in the last decade [52,53,57,58].

The auxiliary trajectories must be for a range of values of the parameter *c* that prevents the appearance of spurious phenomena, as already stated in another article [55]. The dynamic richness of the auxiliary trajectories of the PPG signals reveals the inherent functional complexity to signal dynamics, to which multiple conveniently coupled physiological subsystems contribute. The coordinated action of these subsystems is responsible for homeostatic regulation of the cardiorespiratory system at all times. However, despite the certain global similarity that the auxiliary trajectories of PPG signals may have at first glance, closer scrutiny of each individual shows distinctive signs. These signs could hide more or less diagnostic severe pathologies and, more invariably, the inalienable character of the anatomical and functional configuration of each subject’s cardiorespiratory system.

As far as we know, diffusive dynamics, the cornerstone of the 0–1 test, of a biological signal have never been used to extract biometric characteristics, which gives this work a new operational perspective in physiological biometrics.

### 2.2. Classifier

This paper explores an approach based on convolutional neural networks to identify users through their PPG signals. The proposed system receives two time segments (user A and user B) of PPG signals, each time segmented with three segments of 1000 points each (4 s), as input. The first time segment is the standard segment, and the second time segment is for the user to compare. The system delivers a matching score normalized to the interval [0,1], which defines the degree of agreement between the two incoming PPG segments. If the two input segments belong to the same user, the matching score is closer to 1; if not, the matching score is closer to 0.

#### Architecture

This paper proposes a non-conventional network, as we can see in Figure 2, with an architecture based on a Siamese network whose main trunk is characterized by a fully-connected encoder. It is a multiscale architecture with residual connections according to the guidelines of Szegedy et al. [59]. Fully connected encoder architectures are those traditionally used in classification tasks such as [60,61]. It is well-known for its use in one-shot learning and image verification [62] in the Siamese configuration. To these layers and architectures, somewhat better known in the field, is added a layer to the system that performs preprocessing based on the diffusive behavior peculiar to the PPG signal dynamics [55], highlighted as a new contribution of this paper.

The branch of Siamese network architecture is an Inception-ResNet-V1 [59] due to its recognized capacity as a classifier and its characteristics compared to its previous versions and competing networks:Reduction of architectural bottlenecks [61,63] because the neural network works better if the dimensional input changes are not too drastic. Large dimensional changes can cause a significant loss of information called a “representational bottleneck”.Use of factoring methods to reduce the computational complexity of the convolutions used [63].Use of residual connections between the inputs and outputs of the blocks used [64]. These connections prevent the loss of information and improve the stability of the gradients when training.Use of batch normalization to immunize the network to some extent against scale changes, reduce training time, and avoid covariance displacement [65].

The basic structure of the proposed system takes the form of a network combining 1D information (PPG signals) and 2D information ((p,q)-planes of PPG segments). This structure contains two distinct phases. The first phase consists of a preprocessing layer based on the characteristic (p,q)-planes of the 0–1 test. This phase will have as input six segments of the PPG signal from two users, three belonging to a registered user Pr1,Pr2,Pr3, and the rest to a candidate user Pc1,Pc2,Pc3, not necessarily different. Once these signal segments enter the 0–1 test preprocessing layer, their signals are featured with this process, and six output matrices are obtained, Ir1,Ir2,Ir3 and Ic1,Ic2,Ic3, which can be represented as an image, I=[I1,I2,I3], representing the patterns corresponding to the PPG signals of those users.

The second phase will use as input these six output matrices obtained in the previous phase, in two matrices with three channels each, since each user has three matrices assigned to him or her. This phase consists of a Siamese network whose architecture is based on [59]. This network will use a single coding branch to process the two input matrices separately, with the same trunk and sharing the same weights. Some coded output features, Fr and Fc, will be obtained for each of the input matrices. Once the features are obtained, a relation function of these characteristics quantifies the error between them and quantifies how similar these users are to each other. This error function represents the L1-norm between the vectors of characteristics previously obtained. Once the L1-norm standard is obtained between the characteristics vectors, these will go through a final fully connected binary classification layer. A sigmoidal activation is used to obtain a final C matching score between 0 and 1, quantifying how similar or different the evaluated users are. The architecture can be observed in detail in Table 1, where the sub-blocks that belong to the original Inception-ResNet architecture can be found in the seminal paper [59].

## 3. Material and Methodology

The database used comes from 40 students between 18 and 30 years old, who are non-regular psychotropic substance, alcohol, or tobacco consumers. The students were selected to participate in a national research study to assess how stress reflects in biological signals [66,67]. Signals were captured from the middle finger of the left hand and sampled at a frequency of 250 Hz [66], with the psychophysiological telemetric system “Rehacor-T” version “Mini” from Medicom MTD Ltd [66].

### 3.1. Preprocessing

In practice, the PPG signal is usually impaired by many common noise sources during the signal acquisition process, such as motion artifacts, sensor movements, breathing, etc., and the discretization error (truncation error) involved in normalizing the input signal amplitudes. A common and direct mechanism to mitigate noise is to submit the PPG signal to a bandpass filter. For filtered PPG signals, it uses a Butterworth bandpass filter tuned to different cutoff frequencies. Anything below 0.5 Hz can be attributed to baseline wandering, while anything above 8 Hz is high-frequency noise [68], though some studies have reported clinical information up to 15 Hz [16,69]. To examine the impact that this early preprocessing has on the learning and the final performances of our biometric system, it studies the following variations:Raw data: in this first mode, the PPG signals are not preprocessed and transferred directly, as they were acquired, to the 0–1 test preprocessing layer (see Figure 2), where once segmented, they convert to diffusive geometric maps.Filtered data [0.1–8 Hz]: in this second mode, the PPG signals, before moving to the 0–1 test preprocessing layer, are filtered with a Butterworth bandpass filter with cutoff frequencies at 0.1 and 8 Hz, and the amplitudes are not normalized.Filtered data [0.5–8 Hz]: in this third mode, the PPG signals, before moving to the 0–1 test preprocessing layer, are filtered with a Butterworth bandpass filter with cutoff frequencies at 0.5 and 8 Hz, and the amplitudes are not normalized.Filtered data [0.5–8 Hz] and normalized: in the latter mode, the PPG signals, before moving to the 0–1 test preprocessing layer, are filtered with a Butterworth bandpass filter with cutoff frequencies at 0.5 and 8 Hz, and the amplitudes normalized to the [0,1] interval.

### 3.2. Training

The data used for training are PPG signals obtained for 10 minutes from different individuals with a sampling frequency of 250 Hz in all of them. Each signal is separated into 150 randomly chosen segments (4 s each, which means a 1000-point segment). Each segment generates an image with the 0–1 test. If a database of 40 individuals is used, there are 6000 different PPG segments with all users, and taking three images per user results in 6000360003 possible training combinations. All PPG segments are divided into training, validation, and test sets, composed of 60%, 20%, and 20%, respectively, of the database’s data. Division ranges commonly are chosen to ensure that almost half of the data are used for evaluation.

The problem to be solved by this system is a binary classification problem with only two possible classes: class 0 indicates that the input PPG segments of branch A and branch B do not belong to the same user; class 1 indicates that these segments belong to the same user. Each of the predefined training segments, generated with a specific output label, links these input segments A and B to an output classification, allowing the system to learn how to differentiate or associate the input segments of different users. Once in the training process, a random batch generator will be used, allowing 3 PPG signal segments belonging to user A from among the 40 PPG signals used and another 3 PPG signal segments belonging to user B to be chosen, once again randomized, so that if these two users coincide, an output label will be applied with class 1. At the same time, if not, it will be associated with class 0. This generator allows guaranteeing the highest possible variability, greatly enriching the training and providing it with generality. Once the batches generate, an *Adam* optimizer is used to train the system to recognize similar users.

### 3.3. Optimizer

The used optimizer is *Adam* or adaptive moment estimation [70]. This optimizer is an excellent alternative to the conventional stochastic gradient descent (SGD). It combines the advantages of two previous alternatives [71,72], creating a new approach that uses the averages of the first and second moments of the gradient to adapt the learning rate dynamically.

The training ratio parameter, which indicates the learning rate—how much and how fast the system learns in each period—is crucial and can produce great learning problems if it does not choose correctly. A very high learning rate can produce divergence in training, while a meager rate can easily fall into local training minima or take a long time to complete. When we talk about *Adam*’s adaptive capability, we mean that it starts with a user-defined learning rate, and after, it modifies the learning rate through unsupervised training. This capability allows using an adaptive training ratio that depends strongly on the batch size and how noisy the input is. The training ratio initially used is 10−4.

In addition to *Adam*’s functionality, a callback called early stopping is employed in this training. This tool allows the best weight settings to be saved that the system has achieved throughout the training. In order to achieve this, the training session uses the validation metrics and losses obtained after evaluating the model in each period to save the better-trained weights of the training and avoid undesired effects, such as overfitting. We have to recall that the training sessions were carried out using 100 epochs and a batch size of 5 samples. However, with a predefined number of epochs used, as we have commented before, the early stopping will keep the best of them. The total training time on a GPU NVIDIA GeForce GTX 1080 was 9 h.

### 3.4. Loss Function

The proposed convolutional neural network uses as input two PPG signal segments Ia and Ib, while as output, it uses a binary classification vector C. This binary classification task’s proposed loss function is the cross-entropy (CE), as indicated in Equation (Equation 3), which evaluates the differences between *ground truth* and predictions to provide an output score associated with the input signals’ similarity. In classical machine learning, this loss function has been widely used to solve the problems associated with a binary classification between distributions, d(x) being the correct distribution and d^(x) the estimated one, in such a way that it allows a similarity score for those distributions to be associated.
(3)CEd,d^=−∑∀xd(x)logd^(x).

Binary cross-entropy measures the classifier’s capacity understudy, whose output is a classification level that associates the input with the distribution of interest. The more this classification level decreases, the more the cross-entropy losses increase. The perfect classifier would have zero cross-entropy with a maximum classification level. Usually, this loss function is used in neural networks accompanied by an output activation according to it. In binary cross-entropy, the activation is a sigmoid function, which places the output score level in the interval [0,1], with a smooth transition.

### 3.5. Metrics

Once the modalities in which the experimentation will be carried out are fixed, the metrics used to evaluate the proposed system’s performance are explained:**Precision-Recall curve**. The precision-recall curve depicts the precision vs. the sensitivity (recall) for different operating points (matching score or threshold values). The closer the curve is to the upper right corner (the area under the curve is closer to 1), the more precise and sensitive the system behaves. The accuracy evaluates how often the output is correct (positive). An accurate system is very finicky, validating a legitimate user, i.e., in an accurate system, it is unlikely that an intrusive user will be admitted as valid, but it is also possible that legitimate users will be rejected (false negatives). Sensitivity assesses how permissive the system is, i.e., in a highly sensitive system, it is improbable that a valid user will be rejected, but it is also possible that unregistered users will be admitted as valid (false positives).**ROC (Receiver Operating Characteristic) curve**. The ROC curve depicts sensitivity vs. FPR (false positive rate). The closer the curve is to the upper left corner (the area under the curve is closer to 1), the more sensitive the system behaves without increasing FPR. In short, the ROC curve graphically represents TPR (true positive rate) vs. FPR (false positive rate) for different operating points (matching score or threshold values).**F1 score–threshold curve**. The F1 score–threshold curve complements the information provided by the precision-recall curve. F1 score is a joint and overall metric that brings together the precision and recall values in a unique metric (precision and recall harmonic mean) that allows us to estimate the stability of the system’s performance for different threshold values. In a stable and high-performance system, the range of threshold values for which the curve remains almost constant and close to 1 is virtually a flat line over the whole range.**Equal Error Rate (EER)**. The equal error rate or crossover error rate (CER) is a metric concerning biometric authentication systems that determines a working threshold where FPR (false positive rate) and FNR (false negative rate) are the same. The point where these decision errors cross defines the working point, and the lower the crossover rate, the higher the system’s accuracy. At the experimental level, EER is used as a metric to compare different biometric authentication techniques.

Usually, a high decision threshold identifies an accurate model with a very low FPR (false positive rate); a low threshold value indicates a high sensitivity (too permissive, with a very low FNR (false negative rate)). The precision-recall and ROC curves help us to find the equilibrium threshold. In our case, the criteria for selecting the optimal threshold comes from the EER, but the F1 score–threshold curve tells us if variations of the optimal threshold upwards or downwards would dramatically affect the system performance. Based on the results we will see later, the precision-recall and ROC curves’ equilibrium threshold would not be so critical, as the system’s stability has a wide operating margin for a not insignificant range of working thresholds.

## 4. Results and Discussion

In this section, we show the biometric potential of the diffusive dynamics of the PPG signal. To do this, we explore its operational feasibility under different experimental conditions to mimic its effectiveness in possible real-life scenarios. As an authentication mechanism [5], the biometric architecture consists of two stages: in the first phase, *the enrollment phase*, 12 s of the PPG signal is acquired from each individual using a pulse oximeter. These signal fractions are preprocessed to obtain several (p,q)-planes representative of each subject, and the PPG signal’s diffusive behavior is obtained from the 0–1 test as the biometric pattern. >From these (p,q)-planes, the neural network extracts 51,200 characteristics that encapsulate each individual’s biometric pattern and conveniently stores them in memory. Afterward, in *the verification phase*, a 12 s PPG signal is acquired from anyone who wishes to verify their identity, proceeding to their preprocessing. A classifier and their comparison with the rest of the registered biometric patterns authenticate the identity of the user that requests it. The use of 12 s of a PPG signal in each of the phases of the system is because it is the time necessary to obtain three consecutive segments of the PPG signal (4 s or 1000 points each), with their respective (p,q)-planes from the user, to be recorded or verified. Additionally, 12 s to verify a user’s identity enables applying this system in real environments, since, with this relatively short time, it achieves an accuracy above 90%.

### 4.1. Experimental Conditions

We present two different modalities of experiments that differ in how the database of the PPG signal from various individuals is used for training. We use the whole signal in the first modality, with randomized segments, from 60% of users for training and the remaining 40% for testing. This approximation allows us to show the system’s generalization capacity, with better applicability to real systems, showing its results in new user patterns isolated from the trained users.

The second modality, the most used in the published biometry papers [5,19,43,44,45,46,48,51], uses 60% of all data, with the segments randomly taken between and from all users, for training and 40% for testing, and this means that the used patterns are isolated but belong to the same users, which leads, to a certain extent, to the presence of similarities.

#### 4.1.1. Leaving 40% of Users out of Training

In this first experiment, the training set is 60% of users, and the testing set is the other 40% of users. In this way, the network is trained with 24 users and tested with 16 users never seen before. This experiment allows us to completely isolate 16 users so that the network has never seen a similar pattern in the training phase. Therefore, the register of authorized users does not record the biometrics ID of the 16 users who are kept out.

Figure 3 shows the different EERs for all the input PPG signal modalities used (Section 3.1). For raw and filtered data in the range of 0.1 to 8 Hz, the network’s discriminating power is penalized by the noise present in the signal, distorting and blurring the diffusive geometrical patterns in the (p,q)-planes. As filtering narrows its bandpass in the range of 0.5 to 8 Hz, the impact of noise is attenuated, and the diffusive geometric pattern becomes clearer, allowing the network to discriminate between different users’ biometric patterns more easily. Additionally, if the PPG signal is normalized to the [0,1] interval, once filtered in the range of 0.5 to 8 Hz, the EER has a slight reduction. This effect is because the signal’s normalization improves the numerical quantification, and the diffusive geometric patterns trace a better structural resolution, making it easier to extract the biometric features.

From the EER curve, the working points for each of the preprocessing modes can be measured. These working points can obtain other performance measures, as shown in Figure 4a–c. For raw data and filtered data in the range of 0.1 to 8 Hz, the functional efficiency curves, precision-recall, ROC, and F1 score–threshold curves, behave quite similarly. However, the filtering in the range of 0.5 to 8 Hz, as illustrated in Figure 4a–c, provides a significant enhancement in the system operating performance, especially concerning the stability of the working point, pointed out by the F1 score–threshold curve, which is much higher than the raw data and filtered data in the range of 0.1 to 8 Hz. Unlike in terms of the EER curve in functional efficiency curves, the benefit of [0,1] interval normalization, once filtering the data in the range of 0.5 to 8 Hz, is remarkable. On the one hand, there is a marked improvement in performance for high thresholds, and, on the other hand, in the F1 score–threshold curve, the working point is much more stable than in any other mode.

Table 2 shows the performance metrics of the experiment whereby 40% of the users are left out of training.

#### 4.1.2. Leaving 40% of Data out of Training

In the second experiment, the training set is 60% of the total data, including all users and all users’ segments. The testing set is with the remaining 40% of the data, which means that the network handles (p,q)-planes for all users in the training phase but in a different way than they will be treated for testing, even though they are undoubtedly related to the specific users’ biometric patterns.

This experimental framework establishes a particular environment where the registered users’ database is known and new user registrations are not contemplated. All users are well known to the network as they have previously registered.

Figure 5 shows the different EERs for all the input PPG signal modalities used (Section 3.1). For raw and filtered data in the range of 0.1 to 8 Hz, the network’s discriminating power is similar to that obtained in the preceding experimental framework (Section 4.1.1, Figure 3). The noise present in the signal, which distorts and blurs the diffusive geometrical patterns in the (p,q)-planes, is a critical constraint on the biometrics system’s operational capability.

Nevertheless, contrary to what appears in Figure 3, for filtering in the range of 0.5 to 8 Hz, the network offers high efficiency, with a significant reduction of EER. If, in addition to PPG signal normalization in the interval [0, 1], it applies a filter in the range of 0.5 to 8 Hz, it reaches the lowest EER, very close to zero (6%, as indicated in Table 3). With such credentials, it is clear how proper preprocessing of incoming PPG signals can positively influence the ultimate performance of the biometric system.

From the EER curve, the working points for each of the preprocessing modes can be measured. These working points can obtain other performance measures, as shown in Figure 6a–c. For raw and filtered data in the range of 0.1 to 8 Hz, the functional efficiency curves (precision-recall, ROC, and F1 score–threshold curves) behave almost identical for classification purposes. Otherwise, when filtering in the range of 0.5 to 8 Hz is applied to the input data, a qualitative leap obtains in terms of operational performance, notably about the smooth stability of the F1 score–threshold curve (see Figure 6c). Additionally, normalizing the data to the [0,1] interval, once filtering the data in the range of 0.5 to 8 Hz, enables the network to operate with a quasi-optimal behavior similar to a perfect classifier.

Table 3 shows the experiment’s performance metrics, whereby 40% of the data are left out of training. Finally, we compare in Table 4 performance metrics with other PPG-based biometric methods to consolidate the potential viability attributable to our biometric authentication system. Ratings shown in Table 4 are merely indicative and are limited to the achievements obtained in different experimental scenarios and with different databases. Unfortunately, there is no common roadmap available for the different PPG-based methods to communicate the obtained results. However, always with the utmost respect for the work carried out by the authors, we chose to report the best performances when there is not enough information available to conduct a comparison that is as fair as possible on equal terms.

As Spachos et al. noted [45], the performance of PPG signal acquisition equipment and the environmental conditions when acquiring the signals impact any biometric authentication system’s operational feasibility. So far, most PPG-based biometric systems, as listed in Table 4, extract the representative features of an individual from the morphology of the PPG signal, either directly from the acquired PPG signal itself or with time or frequency domain transformations. Accordingly, the vulnerability of the morphology of the PPG signal to the physical state of the subject and the environmental and instrumental conditions in the signal acquisition process restrict its field of application to biometric environments where very stable conditions are guaranteed, namely when PPG signals, in enrollment and testing phases, were collected under a controlled environment and with accurate sensors.

In light of the above, the inherent biometric limitations of PPG signal morphology are not reflected in the methods collected in Table 4, where an in-depth analysis reveals the high variability experienced by the parameter EER, degrading its expectations, a priori of the most promising, when PPG signals are acquired under different conditions.

Thus, in Yang et al. [50], the best EER is 2.36 with a maximum rank-1 accuracy of 99.69%, evaluated on different datasets, but 10 minutes of PPG signal is required, against the 12 s of our approach. Moreover, as a reference, in Yang et al. [50] the internal computation time for the authentication process, once the data is stored, is about 27 ms, while it is 10 ms in our system. In Yang et al. [49], with 8 min of PPG signal required, a maximum rank-1 accuracy of 99.92% is achieved on three different datasets, but at the expense of an internal computation time for the authentication process of 0.44 s. In Lee et al. [47], the maximum rank-1 accuracy is 99%, evaluated on a dataset containing 42 PPG signals, with roughly 2.5 min of PPG signal required. Either way, all three proposals do not show analysis on different time lapses or different states. In this connection, in Sancho et al. [5], the range of percentage variation of EER is 13.9 (from 6.9 to 20.8%) when evaluated on different time lapses. In Yadav et al. [19], the mean EER is 2.82, evaluated on different states and datasets, or in Spachos et al. [45], it is 12.75, evaluated on different datasets. In the other methods, only the method’s potential is evaluated focusing on the research approach, rather than as a feasible real biometric solution, such as in Karimian et al. [48], where the proposed solution provides an error rate and rank-1 accuracy of 3.91% and 99.44%, respectively, but 8 min of PPG signal is required, against the 12 s of our approach. Either way, and because all of them use PPG individual cycles, exogenous and endogenous factors in the PPG signal’s morphological fluctuations may discourage its use in wearable biometric systems, as consistent and reliable results with proper operations could not be guaranteed. Our approach holds the best EER of all methods, with a 17% margin over the second-best result [50]. Our method is fifth in precision, the best being that obtained in the report of Yang et al. [49]. Finally, in terms of acquisition and processing time, from all the available time values reported by the studies, our method holds first place with 12.01 s. In this sense, it is worth highlighting that our approach does not require new training every time a new user registers; only the user’s template pattern to register is needed, which only takes 12 s to record.

The present proposal opens up a new line of work in PPG-based biometry. Studying its diffusion dynamics replaces the analysis of the PPG signal’s morphology, our (p,q)-planes, which are highly dependent on the vascular bed’s biostructure, an intricate network of tiny blood vessels that branches through body tissues. While deteriorating with age and/or with certain cardiovascular diseases, this vascular microstructure is unique to each individual and maintains a reasonably regular and stable diffusive conductivity over time, making this an excellent biometric marker. Preliminary trials with our biometric authentication system yielded similar performance ratings, with EER and rank-1 accuracy, with one attempt, in the range of about 6% and 97%, respectively, when users, initially registered in a relaxed state, were successfully identified about 30 days later under stress-induced conditions.

## 5. Conclusions

Over the past ten years, the easily accessible PPG signal has attracted those involved in biometric security. Most PPG-based biometric solutions define the biometric signature out of certain features of the PPG signal morphology. Nevertheless, the high variability of the PPG signal morphology, in reaction to changes in measurement conditions and the individual’s psychophysical state, is hampering its adoption as a biometric solution in wearable devices.

In this research work, still in progress, we propose a robust PPG-based biometric authentication system based on the diffusive dynamics of the PPG signal, arguably very stable in changing environments, instead of morphological aspects of the signal. Our biometrics approach is based upon Siamese convolutional neural networks, easily integrated into embedded environments that can reach high speeds in the identification process. An error rate, rank-1 accuracy, and enrollment time of 2%, 97%, and 12 s, respectively, make our proposal the best among the 11 compared state-of-the-art methods in terms of EER and processing time and the fifth-best proposal in terms of rank-1 accuracy, indicating a great significance and potential viability as a real-world biometric system.

With an enrollment time of 12 s, we truly feel that our technical approach can become a real low-cost technological solution. Built-in in miniaturized tensor processing units (TPUs) can be customized for particular use in wearable biometric systems, since once the network has been suitably trained, the authentication methodology does not require successive retraining for reliable serving. Moreover, the memory requirements for storing users’ biometric templates, around 120 kB, pose no apparent constraints on the authorized user database’s portable logistics. With different hardware and software solutions, our efforts aim at reducing PPG signal acquisition time, more in step with the average comparison time, about 10 ms, verifying a user’s biometric credentials requesting access to the system.

Future work involves expanding the dataset with different physiological conditions, but preliminary results with the same individuals under stress conditions and on different days suggest a good operational consistency in the authentication process.

## Figures and Tables

**Figure 1 sensors-21-05661-f001:**
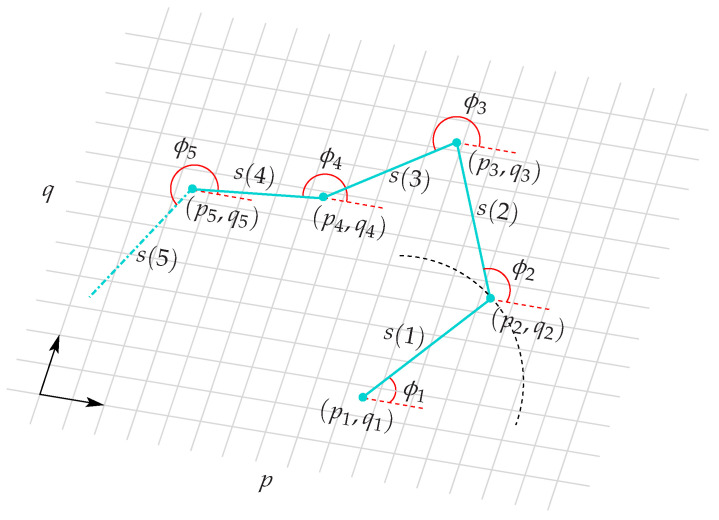
Descriptive construction of the auxiliary trajectory in the (p,q)-plane.

**Figure 2 sensors-21-05661-f002:**
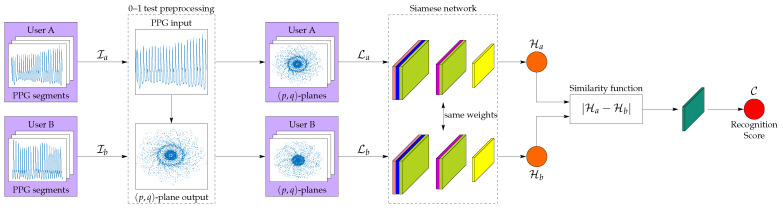
System architecture schematic overview (a zoomed view is shown in Appendix A).

**Figure 3 sensors-21-05661-f003:**
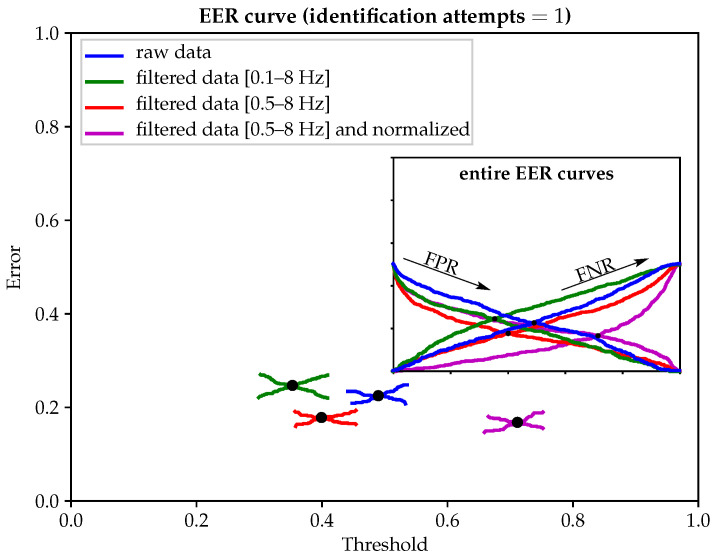
Minimum equal error rate (EER) for different input photoplethysmographic (PPG) signal preprocessing modalities. The inset shows the entire EER curves as well as FPR (false positive rate) and FNR (false negative rate) trends for different threshold values.

**Figure 4 sensors-21-05661-f004:**
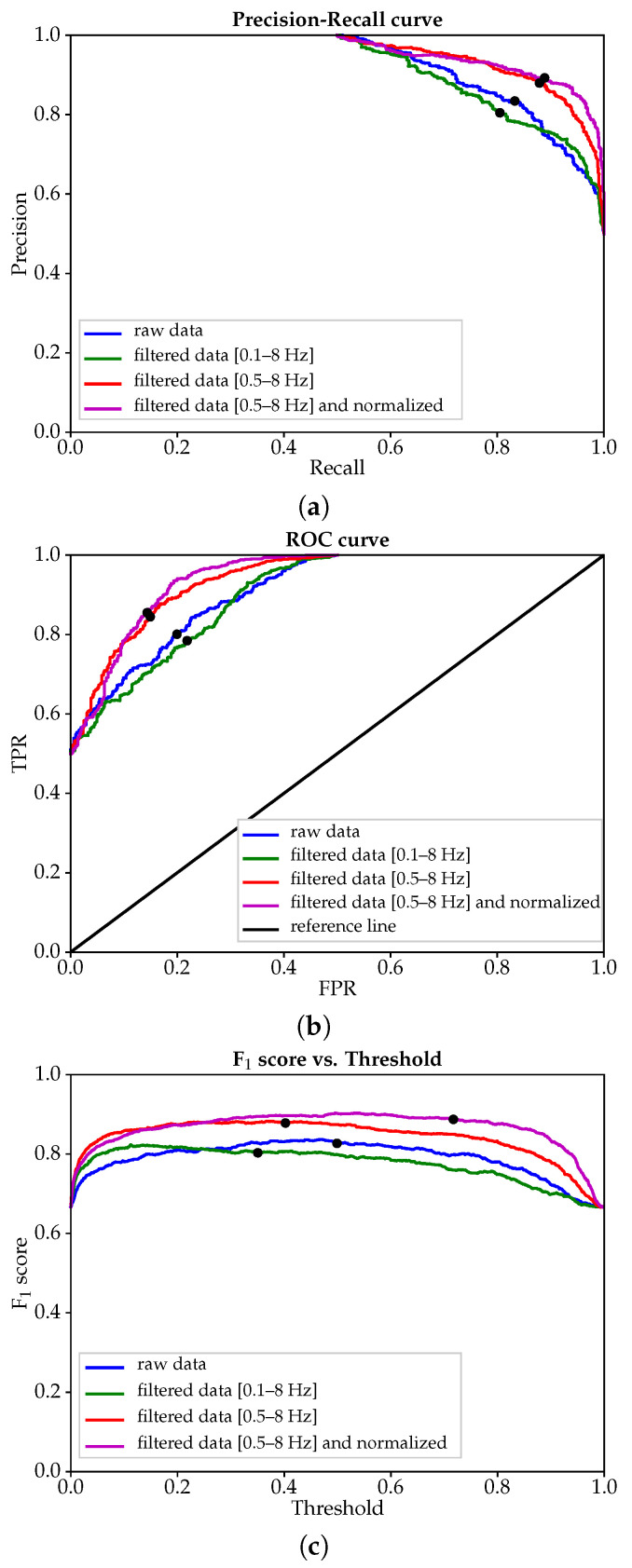
Functional efficiency curves in the case of leaving 40% of users out of training. The working points of the EER curve (see Figure 3) are tagged with the symbol •: (**a**) precision-recall curve; (**b**) ROC curve; (**c**) F1 score–threshold curve.

**Figure 5 sensors-21-05661-f005:**
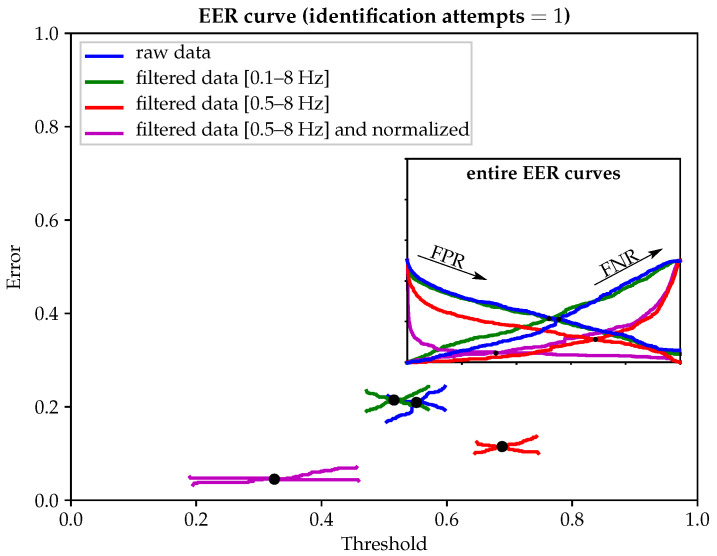
Minimum equal error rate (EER) for different input PPG signal preprocessing modalities. The inset shows the entire EER curves as well as FPR (false positive rate) and FNR (false negative rate) trends for different threshold values.

**Figure 6 sensors-21-05661-f006:**
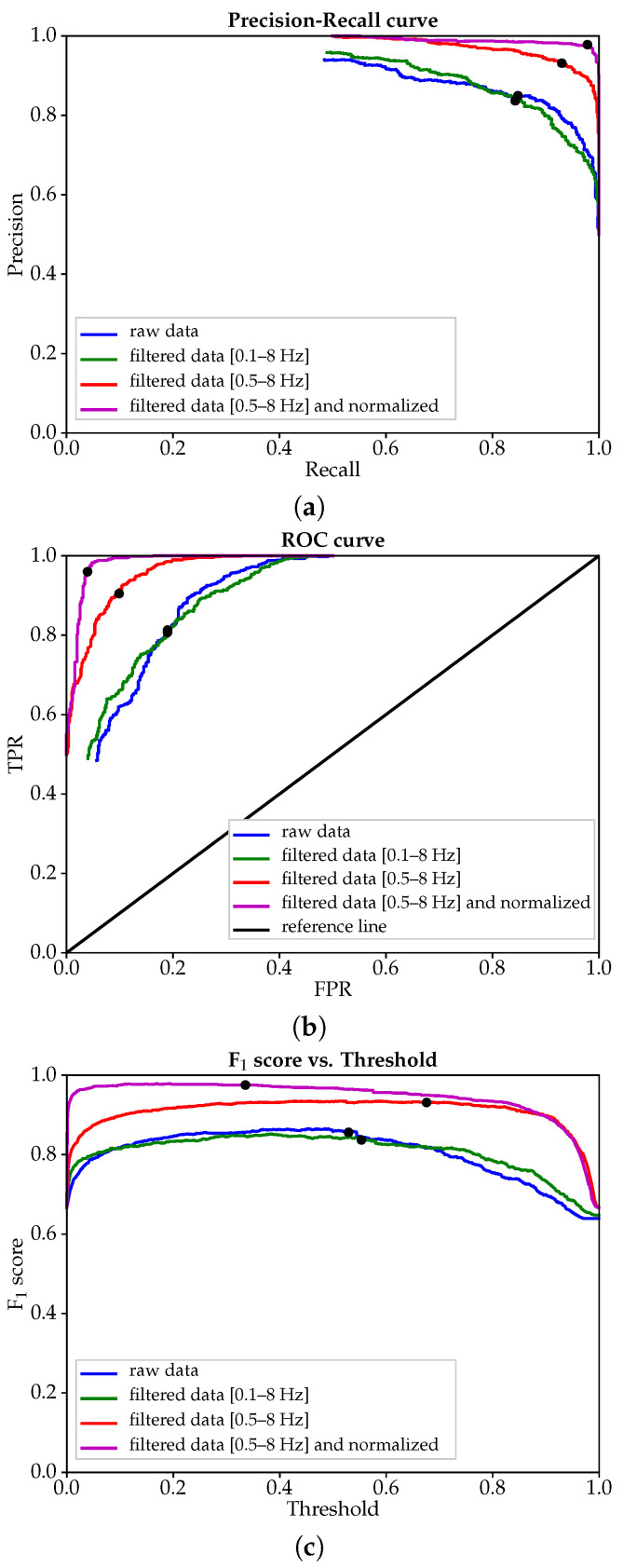
Functional efficiency curves in the case of leaving 40% of data out of training. The working points of the EER curve (see Figure 5) are tagged with the symbol •: (**a**) precision-recall curve; (**b**) ROC curve; (**c**) F1 score–threshold curve.

**Table 1 sensors-21-05661-t001:** Detailed architecture of the proposed CNN (convolutional neural network).

Layer Number	Type	Output Size	Configuration
1A	Input	(1000,3)	—
1B	Input	(1000,3)	—
2	0–1 test preprocessing	2·(299,299,3)	Siamese
3	Stem	2·(35,35,256)	Siamese
4	5× Inception-ResNet-A	2·(35,35,256)	Siamese
5	Reduction-A	2·(17,17,896)	Siamese
6	10× Inception-Resnet-B	2·(17,17,896)	Siamese
7	Reduction-B	2·(8,8,1792)	Siamese
8	5× Inception-Resnet-C	2·(8,8,1792)	Siamese
9	Similarity function	(8,8,1792)	—
11	Flatten	114688	—
12	Dense	1	—
13	Sigmoidal activation	1	—

**Table 2 sensors-21-05661-t002:** Performance metrics for all the input PPG signal modalities used in the case of leaving 40% of users out of training. The thresholds refer to the optimal classification thresholds where EER is minimal for each modality (preprocessing) considered.

Raw Data
**Precision**	**Recall**	**F1 Score**	**Threshold**	**Equal Error Rate (EER)**
0.82	0.82	0.82	0.48	0.22
filtered data [0.1–8 Hz]
0.80	0.80	0.80	0.37	0.23
filtered data [0.5–8 Hz]
0.89	0.89	0.89	0.40	0.19
filtered data [0.5–8 Hz] and normalized in [0,1] interval
0.90	0.90	0.90	0.73	0.18

**Table 3 sensors-21-05661-t003:** Performance metrics for all the input PPG signal modalities used in the case of leaving 40% of data out of training. The thresholds refer to the optimal classification thresholds where EER is minimal for each modality (preprocessing) considered.

Raw Data
**Precision**	**Recall**	**F1 Score**	**Threshold**	**Equal Error Rate (EER)**
0.86	0.86	0.86	0.53	0.21
filtered data [0.1–8 Hz]
0.82	0.82	0.82	0.57	0.22
filtered data [0.5–8 Hz]
0.93	0.93	0.93	0.68	0.11
filtered data [0.5–8 Hz] and normalized in [0,1] interval
0.97	0.97	0.97	0.34	0.06

**Table 4 sensors-21-05661-t004:** The performance of recognition systems based on PPG with state-of-the-art methods compared. Claimed error rates (EERs) involve those in the trial; three attempts were allowed. Acquisition and processing time refers to the system’s time to identify whether the user is valid or not.

PPG-Based Biometric Recognition Method	Equal Error Rate (EER) (%)	Rank-1 Accuracy (%)	Acquisition and Processing Time (s)
Yang et al. 2021 [50]	2.36	99.69	600.027
Yang et al. 2020 [49]	—	99.92	480.44
Lee et al. 2019 [47]	—	99.00	—
Sancho et al. 2018 [5]	6.9	—	21.35
Patil et al. 2018 [51]	23.34	86.67	—
Yadav et al. 2018 [19]	2.82	—	—
Karimian et al. 2017 [48]	3.91	99.44	—
Sarkar et al. 2016 [44]	—	90.53	14.00
Lee and Kim 2015 [46]	3.7	96.04	—
Kavsaoğlu et al. 2014 [43]	—	94.44	13.50
Spachos et al. 2011 [45]	12.75	—	—
Our approach	2.02	97.00	12.01

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
