# Peer review of "Transcending Conventional Biometry Frontiers: Diffusive Dynamics PPG Biometry"

_sensors, 2021, doi:10.3390/s21165661_

Round 1

Reviewer 1 Report

The paper is interesting and well written. It presents the Author's approach to PPG-based biometric system. There are some minor remarks:

  • Figures 2,4,6 should be bigger - they are hardly readable,
  • in line 326 we can read 'real-world PPG' - what does it mean?
  • Literature is vast and appropriate, but there is no element from 2021.

Author Response

Thanks for your helpful comments.

Point 1: Figures 2,4,6 should be bigger - they are hardly readable,

Response 1: An appendix includes a duplicate of Figure 2, so the architecture used is better seen. See pages 6, and 19 on the new manuscript.

To facilitate visualization of Figures 4 and 6, they arrange differently. See pages 12, and 15 on the new manuscript.

 Point 2: in line 326 we can read 'real-world PPG' - what does it mean?

Response 2: The expression "real-world PPG" in the text uses only on that line. We agree with the reviewer that it means nothing; it is redundant, so we remove "real-world" at line 326.

 Point 3: Literature is vast and appropriate, but there is no element from 2021

Response 3: The only references find (years 2019, 2020, and 2021), related only to PPG biometry, reflect in the new table 4. See pages 3, 16, and 17 on the new manuscript.

Reviewer 2 Report

(1)The authors mentioned several signals in Lines 82~88. It is better to use a table to introduce and compare them.
(2)Some characters in figures are too small, such as Figs. 4 and 6.
(3)Some data in Table 4 are missing. Are the results of the methods performed on the same dataset and conditions? If not, the comparisons are unfair.
(4)The methods in Table 4 were published before 2019. Are there any other state-of-the-art methods that can be compared with and published after 2018?

Author Response

Thanks for your helpful comments.

 Point 1: The authors mentioned several signals in Lines 82~88. It is better to use a table to introduce and compare them.

Response 1: This paragraph only intends to highlight the increasingly prominent role of biological signals in the field of biometrics. It is not our intention to establish a comparison between the results obtained with the PPG signal and those that could derive from other biological signals.

Point 2: Some characters in figures are too small, such as Figs. 4 and 6.

Response 2: To facilitate visualization of Figures 4 and 6, they arrange differently. See pages 12, and 15 on the new manuscript.

Point 3:  Some data in Table 4 are missing. Are the results of the methods performed on the same dataset and conditions? If not, the comparisons are unfair.

Response 3:  The published data on PPG biometrics are compiling and summarized in Table 4 to have a reference criterion on the efficiency of our proposal.

Our objective is to corroborate that the diffusive behavior of the PPG signal constitutes a promising biometric pathway, as the results show.  Indeed, the data of the other proposals come from different databases, and, in fact, the next step, as indicated in the conclusions, would be to extend the study to other databases and to other experimental protocols that allow evaluating the robustness of the dynamic variant that we propose.

In the context of PPG signal biometrics, there is no standard framework for rigorously evaluating the efficiency of a biometric system. As far as we know, the first steps are still taking, and the different proposals do not always attend to the EER parameters and Rank-1 accuracy as reliability indicators, as they tacitly use since the analysis's appearance fingerprints. Few proposals evaluate the time the authentication process takes, and it is a critical indicator for its implementation in a real device.

The data included in Table 4 it is the data available in each reference. The missing data is because the authors did not give it. Some comments on it is done on the text.

Point 4: The methods in Table 4 were published before 2019. Are there any other state-of-the-art methods that can be compared with and published after 2018?

Response 4: The only references find (years 2019, 2020, and 2021) reflect in the new table 4. These references are only on PPG biometry. We do not study solutions using more than one biological signal. See pages 3, 16, and 17 on the new manuscript.

Reviewer 3 Report

This paper is written on biometric authentication systems using PPG, and is very interesting.

It is possible that I am mistaken, but I have doubts about the following contents.

Biometrics should use data  (PPG in this paper)  measured in various states of the same subject to evaluate its performance.
As the authors point out, PPG is affected by a variety of artifacts.
The shape of the waveform also changes depending on the position of the sensor and the pressure applied to the fingertip.
Please perform two or more measurements on the same subject and evaluate some of them separately as supervised data and test data.
If the above evaluation has been done, please specify it in 3. Material and Methodology.

Author Response

Thanks for your helpful comments.

Point 1: Biometrics should use data (PPG in this paper) measured in various states of the same subject to evaluate its performance.

Response 1: We agree with the reviewer.

This comment is related to the reason of indicating, at the end of section 4. Results and discussion, the sentence (number lines correspond to the revised version):

  ... Preliminary trials

585 with our biometric authentication system yielded similar performance ratings, with EER

586 and rank-1 accuracy, with one attempt, in the range of about 6% and 97%, respectively,

587 when users, initially registered in a relaxed state, were successfully identified about 30

588 days later under stress-induced conditions.

Remember data is from (number lines correspond to the revised version):

... The students were selected to

299 participate in a national research study to assess how stress reflects in biological signals

300 [64,65].

at the beginning of section 3. Material and Methodology.

Point 2: As the authors point out, PPG is affected by a variety of artifacts.

The shape of the waveform also changes depending on the position of the sensor and the pressure applied to the fingertip.

Response 2: Indeed, the idea that it raises is interesting. However, it is unfortunately not possible to address it in this work, since we only have the PPG signals that the study directors provided us, and our purpose was to evaluate the degree of effectiveness of this novel biometric approach for the sake of future implementation in a low cost, fast and reliable device.

We believe this new biometric variant, where the morphological characteristics are not directly evaluated but rather the PPG signal's diffusive behavior, is more robust against artifacts derived from the measurement procedure (involuntary movements, location, and anchoring or adjustment (coupling) sensor, etc.). Apart from the changes that physical-psychic states already introduce overtime in the waveform of the PPG signal.

Point 3: Please perform two or more measurements on the same subject and evaluate some of them separately as supervised data and test data.
If the above evaluation has been done, please specify it in 3. Material and Methodology.

Response 3:  Related to the changes that physical-psychic states introduce over time in the waveform of the PPG signal we refer to in the study, preliminary tests are promising since temporary changes are not affecting the recognition system, an aspect that, in other cases, proposals are generating reliability problems in user recognition.

See Response 1.

These trials will follow by the PPG signal data provided by other databases, to which other experimental protocols would incorporate different psychological states, cardiorespiratory diseases, stress, ..., as well as the "procedural errors" suggested by the reviewer, would be added, and that could pose a risk to take into account in the authentication process. Be that as it may, the reviewer's suggestion is valuable and will consider in the planned experimentations. Thanks a lot.

Round 2

Reviewer 2 Report

The current version is better than the previous one.
(1)The authors could read and cite “Editorial: Special issue on advanced biometrics with deep learning” (2020), in which many related state-of-the-art methods are reviewed.
(2)Please check all the typos and details to improve the writing.

Author Response

Point 1: The authors could read and cite “Editorial: Special issue on advanced biometrics with deep learning” (2020), in which many related state-of-the-art methods are reviewed.

Response 1: We added the reference recommend (in the text on lines 73-74 of the new version of the article, highlighted in yellow). Indeed, the new deep learning models will provide a clearer vision of interpreting the learning mechanisms in the pattern recognition process. In this way, the models can be simplified, alleviating their operational complexity and the computational load, the latter a critical factor in portable devices, such as the ubiquitous mobiles. Thanks a lot for the suggestion.

Point 2: Please check all the typos and details to improve the writing.

Response 2: The entire text is rechecked.

Reviewer 3 Report

The authors respond appropriately to peer-reviewed comments.

I am also looking forward to a new paper based on the new experimental design.

Author Response

Thanks a lot for your good wishes and your interest in our investigations. We know that it is a slow procedure to maintain the rigor required by a properly executed protocol. In any case, we are convinced that we will obtain convincing results. More importantly, we will draw interesting conclusions about the behavior of the PPG signal concerning the physiological system from which it derives.